# Determinants of Immunovirological Response among Children and Adolescents Living with HIV-1 in the Central Region of Cameroon

**DOI:** 10.3390/tropicalmed9020048

**Published:** 2024-02-14

**Authors:** Rodolphe Steven Dobseu Soudebto, Joseph Fokam, Nelly Kamgaing, Nadine Fainguem, Ezechiel Ngoufack Jagni Semengue, Michel Carlos Tommo Tchouaket, Rachel Kamgaing, Aubin Nanfack, Yagai Bouba, Junie Yimga, Collins Chenwi Ambe, Hyacinthe Gouissi, Jeremiah Efakika Gabisa, Krystel Nnomo Zam, Alex Durand Nka, Samuel Martin Sosso, Gregory-Edie Halle-Ekane, Marie-Claire Okomo, Alexis Ndjolo

**Affiliations:** 1“Chantal BIYA” International Reference Center for Research on HIV/AIDS Prevention and Management (CIRCB), Yaoundé P.O. Box 3077, Cameroon; dobseusteven94@gmail.com (R.S.D.S.); kmnelly2006@yahoo.fr (N.K.); fainguem_dine@yahoo.fr (N.F.); ezechiel.semengue@gmail.com (E.N.J.S.); tommomichel@yahoo.fr (M.C.T.T.); r.kamgaing@yahoo.it (R.K.); a_nanfack@yahoo.fr (A.N.); junie_flore@yahoo.fr (J.Y.); collinschen@yahoo.co.uk (C.C.A.); davygouissi@gmail.com (H.G.); efakikagabisa@gmail.com (J.E.G.); mknnomozam@gmail.com (K.N.Z.); nkaalexdurand@yahoo.com (A.D.N.); martinsosso@yahoo.it (S.M.S.); andjolo@yahoo.com (A.N.); 2Faculty of Medicine and Biomedical Sciences (FMSB), University of Yaoundé I, Yaoundé P.O. Box 3077, Cameroon; okomo2015@gmail.com; 3Faculty of Health Sciences, University of Buea, Buea P.O. Box 63, Cameroon; halle-ekane.edie@ubuea.cm; 4Department of Gynecology Obstetrics, University Teaching Hospital (CHU), Yaoundé P.O. Box 3077, Cameroon; 5Microbiology and Clinical Microbiology, UniCamillus—Saint Camillus International University of Health Sciences, 00131 Rome, Italy; romeobouba@yahoo.fr; 6Department of Experimental Medicine, Faculty of Medicine and Surgery, University of Rome “Tor Vergata”, Via Montpellier 1, 00133 Rome, Italy; 7National Public Health Laboratory, Yaoundé P.O. Box 3077, Cameroon

**Keywords:** HIV, immunological failure, viral suppression, paediatrics, Cameroon

## Abstract

About 90% of new HIV-1 infections in children occur in sub-Saharan Africa, where treatment monitoring remains suboptimal. We sought to ascertain factors associated with immunovirological responses among an ART-experienced paediatric population in Cameroon. A laboratory-based and analytical study was conducted from January 2017 throughout December 2020 wherein plasma viral load (PVL) analyses and CD4 cell counts were performed. Viral suppression (VS) was defined as PVL < 1000 copies/mL and immunological failure (IF) as CD4 < 500 cells/µL for participants ≤5 years and CD4 < 250 cells/µL for those >5 years; *p* < 0.05 was considered statistically significant. Overall, 272 participants were enrolled (median age: 13 [9–15.5] years; 54% males); median ART duration 7 [3–10] years. Globally, VS was achieved in 54.41%. VS was 56.96% in urban versus 40.48% in rural areas (*p* = 0.04). IF was 22.43%, with 15.79% among participants ≤5 years and 22.92% among those >5 years (*p* = 0.66). IF was 20.43% in urban versus 33.33% in rural areas (*p* = 0.10). Following ART, IF was 25.82% on first-line (non-nucleoside reverse transcriptase inhibitors; NNRTI-based) versus 10.17% on second-line (protease inhibitor-based) regimens (*p* = 0.01). Interestingly, IF was 7.43% among virally suppressed versus 40.32% among virally unsuppressed participants (*p* < 0.0001). A low VS indicates major challenges in achieving AIDS’ elimination in this paediatric population, especially in rural settings and poor immune statuses. Scaling up NNRTI-sparing regimens alongside close monitoring would ensure optimal therapeutic outcomes.

## 1. Introduction

The Human Immunodeficiency Virus (HIV) affects the immune system by infecting T lymphocytes and specifically the clusters of differentiation 4 (TCD4), resulting in the creation of viral factories for the production of new virions [1]. HIV infection is immunocompromising, resulting in the Acquired Immune Deficiency Syndrome (AIDS). In the absence of treatment, HIV infection progresses gradually and can be monitored using two biomarkers that evolve inversely: the number of T CD4 and the plasma viral load (VL) [2]. The management of people living with HIV was radically changed in 1996 with the introduction of highly active antiretroviral therapy (HAART) [2]. This therapy does not cure the infection but suppresses viral replication in the body, prolonging the life of people living with HIV, making the plasma VL undetectable (<50 copies/mL), restoring the number of T CD4 (≥500 cells/µL) while preserving immunological functions, preventing the emergence of resistance mutations, and reducing transmission [2]. However, while HIV viral load responds immediately to antiretroviral therapy (ART), dropping to undetectable typically in a few weeks, in the case of CD4 count the response is slower. If the CD4 count is low when starting ART, it may take months or more for it to climb to 500 cells/µL. As a result of antiretroviral treatment, HIV/AIDS has thus become a chronic disease in Europe and in African countries where these drugs are available [3].

In 2021, the United Nations Organization on Human Immunodeficiency Virus/Acquired Immune Deficiency Syndrome (UNAIDS) estimated the number of people living with HIV (PLHIV) worldwide at 38.4 million. Sub-Saharan Africa remains the most affected area with 70% of PLHIV, i.e., 25.6 million people [4]. An estimated 1.8 million adolescents aged 10 to 19 years are infected with HIV worldwide, 90% of whom live in sub-Saharan Africa [5]. In sub-Saharan Africa, 90% of HIV infections in children are transmitted by the mother [6]. Nevertheless, new HIV infections in children since 2010 have decreased by 41%, from 280,000 [190,000–430,000] in 2010 to 160,000 [110,000–260,000] in 2018 [4]. Although the percentage of AIDS-related deaths decreased by 35% globally between 2005 and 2013, the proportion remains high among adolescents living with HIV (approximately 50%) making HIV the second leading cause of death among adolescents globally and the leading cause of death among adolescents in sub-Saharan Africa [7]. This high mortality is largely attributable to difficulties in ensuring appropriate transition from paediatric to adult ART regimens [8].

In Cameroon, HIV prevalence is 2.7% according to the results of the latest demographic and health survey conducted in 2018 [9]. Among adolescents, this prevalence is estimated at 2% [10]. Approximately 224,000 new people received ART, including 8,500 children under 15 years of age, in 2018 [11]. The main challenge in paediatrics remains to minimize virologic failure through drug resistance testing approaches, so that treatment options remain available into adolescence and adulthood [12]. Proper management of pregnant women living with HIV significantly reduces the risk of HIV transmission from mother to child [13]. It should be noted that viral load is measured less frequently in resource-limited countries than in resource-rich ones (usually once a year rather than every three months) [5]. As a result, there is less information on viral suppression and immunological monitoring data in children on ART in resource-limited settings (RLS) [14]. Therefore, given the vulnerability of children and adolescents to HIV infection as compared to adults, monitoring of paediatric virological and immunological response in this tropical setting would be essential for optimizing strategies for an improved clinical management in paediatrics and to inform national policies toward a better life expectancy on ART for this underserved population. In this prospect, we sought to ascertain the determinants of immunological and virological responses among children and adolescents on ART in the Cameroonian context; our findings will be pivotal to assess the burden of immunological failure and evaluate our national progress toward the achievement of the third 95 target of UNAIDS by 2030. Of note, these UNAIDS targets stipulate that 95% of PLHIV should know their status, 95% of those diagnosed should receive ART, and 95% of those on ART should have their viral load suppressed (<1000 copies/mL).

## 2. Materials and Methods

### 2.1. Study Design, Setting, and Populations

This was a laboratory-based and analytical study conducted among HIV-1-infected children and adolescents on antiretroviral treatment from January 2017 throughout December 2020 at the “Chantal BIYA” International Reference Centre for research on HIV/AIDS prevention and management (CIRCB) in Yaoundé, Cameroon. Briefly, the CIRCB is a reference public centre specialized on (a) early HIV diagnosis in infants in the frame of the national prevention of mother-to-child transmission program; (b) diagnosis of co-infections with HIV; (c) viral load measurement; (d) CD4 and CD8 T lymphocyte counts; (e) biochemical and haematological tests for drug follow-up; (f) genotypic resistance testing (GRT) at subsidized costs for the surveillance of HIV drug resistance (HIVDR); with quality control programs conducted in partnership with Quality Assessment and Standardization of Indicators (QASI) and other international organizations (http://www.circb.cm/btc_circb/web/, accessed on 13 February 2024).

HIV-1-infected participants included in this study were consecutively enrolled at their follow-up visit at the CIRCB for research on HIV/AIDS prevention and management based on the following inclusion criteria: (a) aged 0–9 years (children) and 10–19 years (adolescents); (b) on ART for at least 12 months.

### 2.2. Data Collection

The data collection tool was the electronic register and a survey form. The interviews were conducted at the Technical Units of CIRCB where each participant was given an information sheet to help them better understand the purpose of the study. The information collected consisted of sociodemographic data (sex, age, category of health facility, geographical area), clinical data (current treatment regimen, duration of antiretroviral treatment), and biological data (CD4 count and quantification of plasma HIV viral load). Adherence was evaluated during interviews based on patients’ answers.

### 2.3. Procedures

After obtaining the consent or assent of the participants in this study, whole blood was collected (5 mL × 2) from the veins of the patient’s elbow in ethylene diamine tetraacetic acid (EDTA) tubes for CD4 count and quantification of plasma HIV-1 viral load. The biological analyses (plasma for the quantification of the HIV-1 plasma viral load and whole blood for the CD4 count) were carried out for each sample in the medical analysis laboratory of the CIRCB. For HIV-1 plasma viral load quantification, the sample contained in one of the two EDTA tubes was centrifuged at 1500 rpm for 15 min to obtain the plasma. The other EDTA tube was used immediately for the CD4 lymphocyte assay. Samples that were not immediately analysed (quantification of plasma HIV-1 viral load), 800 µL of plasma, were stored below −20 °C until the date of analysis.

#### 2.3.1. Quantification of Plasma HIV-1 Viral Load

Quantification of HIV-1 viral load was performed on plasma samples using real-time Polymerase Chain Reaction (PCR) on the *Abbott m2000rt* HIV-1 platform according to the manufacturer’s instructions (*Abbott RealTime HIV-1,* Des Plaines, IL, USA) based on the following principle: An RNA sequence unrelated to the HIV-1 target sequence is introduced into each sample at the start of sample preparation. This sequence is simultaneously amplified in the RT-PCR and used as an internal control to demonstrate the correct processing of each sample. The amount of HIV-1 target sequence present at each amplification run is measured using fluorescently labelled oligonucleotide probes. The probes do not generate a signal unless they are specifically linked to the amplified product. The amplification cycle during which the fluorescent signal is detected is proportional to the logarithm of the concentration of RNA-1 present in the original sample. Quantification of HIV-1 genomic RNA takes place in several phases. For extraction, 0.6 mL of plasma was loaded into an *Abbott* m2000sp instrument, combined with the *Abbott HIV-1* master mix containing an internal RNA control, primers, and probes targeting the pol-integrase gene. Amplification, was performed using a thermocycler m2000rt after automated extraction and sample preparation using an *Abbott* m2000sp instrument. At the end of the process, the amplification cycle (CN) during which the fluorescent signal is detected is proportional to the logarithm of the concentration of RNA-1 present in the sample. The detection sensitivity for the assay is 40 copies/mL (with a lower and upper detection limit of <40 copies/mL and >10,000,000 copies/mL HIV-1 RNA, respectively). Viral suppression (VS) was defined as viral load (VL) < 1000 RNA copies/mL blood plasma, and very low-level viraemia <50 copies/mL was considered controlled viraemia.

#### 2.3.2. CD4 Lymphocyte Count

CD4 lymphocyte counts were obtained by flow cytometry using the CyFlow counter (Sysmex Corporation, Kobe, Japan), the principle of which is as follows: flow cytometry is the measurement of the properties of a cell population moving in a sheath of liquid. The cells to be analysed must therefore be in suspension. These cells pass one by one through a beam of light (green solid laser) at 532 nm, and the Partec flow cytometer (Kent, United Kingdom) then detects the cell’s ability to diffract the incident light and emit CD4 PE fluorescence. The light scattered and the fluorescence emitted by each cell are captured by the CyFlow counter detectors and transmitted to a computer. The data collected are analysed using CyView™ software and presented in the form of histograms or graphs. According to the manufacturer’s instructions, the whole blood sample (20 µL) is mixed with the reagent (20 µL) containing CD4 mAb PE monoclonal antibody. These fluorochrome-labelled antibodies bind specifically to leukocyte surface antigens. The labelled samples are then treated with a buffer solution (800 µL) that lyses the red blood cells under mild hypotonic conditions while preserving the leukocytes through a light beam (green solid-state laser) at 532 nm. The data collected were analysed using CyView™ software and are presented in the form of histograms or graphs. Immunological failure was defined as CD4 < 500 cells/µL in participants aged 1 to 5 years and CD4 < 250 cells/µL in participants aged over 5 years.

### 2.4. Statistical Analysis

The data collected for this study were entered and processed using Excel 2016 spreadsheet software, and analyses were carried out using EpiInfo version 7 software for the analysis of mono- and bivariate data. Variables are expressed as frequencies; proportions and 95% confidence intervals were calculated, and data were summarized using tables. Hypothesis tests were performed using the chi-square test and Fisher’s exact test to show associations between the different variables. The level of statistical significance was set at *p* < 0.05.

### 2.5. Ethical Considerations

Ethical approval for this study was obtained from the Institutional Review Board of the Faculty of Medicine and Biomedical Sciences of the University of Yaoundé I (260/UY1/FMSB/VDRC/DAASR/CSD). Data collection was carried out after obtaining the research authorization from the general directorate of the “Chantal BIYA” International Reference Centre for research on HIV/AIDS prevention and management. We respected the anonymity of the participants included in this study by arbitrarily assigning them an identification code that was used during the collection of information. Assent forms were obtained from all participants and legal guardians provided their written informed consents.

## 3. Results

### 3.1. Characteristic of the Study Population

A total of 272 participants living with HIV (71 children and 201 adolescents) were recruited for this study as detailed in Table 1. The participants included in this study were 53.68% (146/272) males and 46.32% (126/272) females, with a sex ratio of 1.16 in favour of males. The median (interquartile range, IQR) age of the study population was 13 (9–15.5) years, with 7 (5–8) years for children and 14 (12–16) years for adolescents. The age groups 6 to 9 and 10 to 15 years were the most represented, respectively, in children, 73.24% (52/71), and adolescents, 66.17% (133/201). Of the 272 participants included in this study, most lived in urban areas (84.56%, or 230/272) compared with 15.44% (42/272) in rural areas. By category of health facility, 93.38% (254/272) of the participants were treated in public facilities, compared with 6.62% (18/272) in private facilities. About 61.40% (167/272) of the participants included in this study were diagnosed during prevention of mother-to-child transmission, i.e., 71.83% (51/71) of the children and 57.71% (116/201) of the adolescents. Adherence was routinely measured by self-reporting, and all participants reported being adherent; adherent being defined as having received 95% or above of prescribed ART. The median duration of antiretroviral treatment for the participants in this study was 7 (3–10) years, i.e., 5 (3–6) years for children and 7 (4–11) years for adolescents. The majority of the participants in this study was on first-line therapy (78.31%, 213/272) comprising two nucleoside reverse transcriptase inhibitors (NRTI) and one non-nucleoside reverse transcriptase inhibitor (NNRTI), compared with 21.69% (59/272) on second-line therapy comprising two NRTI and one protease inhibitor (PI). The antiretroviral combinations AZT + 3TC + NVP and TDF + 3TC + EFV were the most represented, respectively, in children (36.62% or 26/71) and adolescents (28.86% or 58/201).

### 3.2. Determinants of Viral Suppression

The median plasma HIV viral load of participants in this study was 415 [<40–36,523] copies/mL, with 302 [<40–29,198] copies/mL in children and 545 [<40–40,543] copies/mL in adolescents. The viral suppression rate (<1000 copies/mL as defined by the World Health Organization; WHO) in the overall population was 54.41%, i.e., 57.75% (41/71) in children and 53.23% (107/201) in adolescents. In contrast, controlled viraemia (plasma HIV viral load <50 copies/mL as defined by the WHO) was 26.76% and 28.86% in children and adolescents, respectively. Table 2 below shows the distribution of plasma viral load in the study population (Table 2a) and determinants of viral suppression (Table 2b).

### 3.3. Factors Associated with Immunological Failure

The median (IQR) CD4 count in this study was 576 (301–893) cells/µL, or 1113 (598–1505) cells/µL in children aged 1–5 years and 566 (295–855) cells/µL in children aged >5 years and adolescents. The immunological failure rate in the overall population was 22.43% (61/272), i.e., 15.79% (3/19) in children aged 1 to 5 years and 22.92% (58/253) in children aged over 5 years and adolescents. Table 3 shows the distribution of CD4 counts of the participants included in this study and the determinants of immunological failure.

## 4. Discussion

This study evaluated the determinants of viral suppression and immune response in a paediatric population on ART for at least 12 months. The M/F sex ratio was 1.16, suggesting a similar distribution of boys and girls within our study population. Our results are consistent with a study conducted by Tadesse et al. who reported 56% of male children in Ethiopia [15], translating similar attendance of young boys and girls to health facilities in resource-limited settings.

The median (IQR) age of the study population was 13 (9–15.5) years, with 7 (5–8) years in children and 14 (12–16) years in adolescents. This result could be explained by the predominance of the age groups 6 to 9 years (in children) and 10 to 15 years (in adolescents). Our results are similar to those of a study conducted by Fokam et al. at the Health Centre of the National Social Security Fund in Yaoundé, Cameroon, which found a median (IQR) age of 13 (11–16) years among adolescents, and the study conducted by Tadesse et al. in Ethiopia, who found a median age of 9 (5–12) years among the population aged 3 to 12 years [8,15].

According to the timing of the HIV diagnosis, 167/272 (61.4%) of the participants in this study were tested during prevention of mother-to-child transmission, i.e., 51/71 (71.83%) and 116/201 (57.71%) children and adolescents, respectively. These results differ from those of Fokam et al. who found that 49.7% of participants were diagnosed following a routine consultation [16]. However, this difference could be explained by the fact that our study population consisted solely of children and adolescents, whereas Fokam et al. included mainly adults.

In our study, the median HIV plasma viral load of the participants was 415 [0–36,523] copies/mL, 302 [0–29,198] copies/mL in children and 545 [0–40,543] copies/mL in adolescents. Overall, 54.41% of subjects were virally suppressed (i.e., HIV plasma viral load <1000 copies/mL as defined by the WHO [17]), i.e., 57.75% (41/71) in children and 53.23% (107/201) in adolescents. On the other hand, the prevalence of patients with controlled viraemia (<50 copies/mL as defined by the WHO) was 26.76% and 28.86% in children and adolescents, respectively. These results show that the participants in this study have a poor virological response. Indeed, previous studies have shown that the low virological response could be justified by the higher viral replication and less effective immune response [18]. The viral suppression rate in this study is below that of a study conducted in South Africa by Teasdale et al. who found a viral suppression of 57.7% [14]. In contrast, a study by Isaac et al. found a viral suppression of 48.70%, 49.70% in children and 47.9% in adolescents [19]. This difference in prevalence between the two groups could be explained by the general characteristics of the population studied and the duration of ART. Importantly, at the moment of the study, all participants reported being adherent to ART. Of note, suboptimal adherence to treatment is known as a major determinant of ART failure among children in Cameroon [16]. In effect, a suboptimal adherence level is a major challenge often reported in children and adolescents due to a lack of psychological support and close monitoring by tutors and clinicians [12,13,14,15,16,17,19]. In our present study, adherence was measured by self-reporting, which is a method with possible recall bias from clients. Furthermore, these children/adolescents had a past history of poor adherence and these observations could have contributed to the present poor rate of viral suppression.

According to geographical location, most of the participants included in this study were from urban areas: 230/272 (84.56%) or 64/71 (90.14%) children and 166/201 (82.59%) adolescents. The viral suppression was 131/230 (56.96%) in the urban locality against 17/42 (40.48%) in the rural locality (*p* = 0.04). This result could be justified by the fact that in rural areas, there are difficulties in accessing health care facilities on a regular basis, recurrent drug stockouts, and a lack of adequate infrastructure and qualified personnel, underlining the urgent need for specific approaches to monitor and manage ART uptake in rural areas [20]. On the other hand, urban areas host several molecular diagnostic laboratories where VL monitoring can be performed, whereas this is not the case in rural areas. Our results are low compared to those reported by Tchouwa et al. in the first national study for estimation of viral suppression, which found a frequency of 75.00% and 67.70%, respectively, in urban and rural areas [20].

According to the category of healthcare facility, 254/272 (93.38%) of the participants in this study were followed up in public centres, compared with 18/272 (6.62%) in private facilities. Viral suppression was 140/254 (55.12%) in public healthcare facilities compared with 8/18 (44.44%) in private facilities. This predominance of participants in public healthcare centres could be justified by the fact that HIV plasma viral load testing is free in Cameroon for all PLHIV that are monitored within the healthcare system.

By gender, viral suppression in the overall study population was higher in the female gender than the male gender (57.94% vs. 51.37%). Our results are in contrast to a nationwide study conducted by CAMPHIA in 2017 that reported similar viral suppression in males and females (80.1% vs. 79.2%) [21]. Several studies have found that the male gender, particularly in adulthood, is more likely to experience virologic failure [22,23]. In effect, studies have shown that males stand a greater risk with the tendency to express their virility through multiple sexual partners, refusal to use condoms, alcohol abuse, and poorer use of health services [16], which leads to poorer adherence and treatment interruptions that promote treatment failure. In addition, it is well known, particularly in sub-Saharan Africa, that men are more likely to die from HIV/AIDS than women, because they are less knowledgeable about HIV/AIDS and tend to present to health services at an advanced stage of the disease [24]. While maintaining and improving women’s access to ARVs and SVs through existing programmes, men must not be left behind. Given that the current design of the healthcare system may be responsible for these gaps, designing separate interventions for men and women, filling gaps in the HIV/AIDS care continuum, and increasing case detection through PMTCT, index-based testing, and workplace testing [22] can help achieve the UNAIDS target of 95%. The participants in this study were on first-line therapy consisting of two NRTI and one NNRTI at a frequency of 78.3%, versus 21.7% on second-line therapy consisting of two NRTIs plus one PI. Lamivudine was administered to all patients as the main component of NRTIs. Children with viral suppression were mostly on first-line therapy with 31/55 (56.36%) versus 10/16 (62.50%) on second-line therapy. In adolescents, it was 89/158 (56.33%) on first-line therapy versus 18/43 (41.86%) on second-line therapy. Among the anti-retroviral combinations, the most prescribed triple therapy was AZT + 3TC + NVP in children (47.27%) and TDF + 3TC + EFV in adolescents (36.71%) on first-line therapy. For second-line therapy, all the children (100%) and 62.79% of the adolescents were on a two NRTIs + LPVr protocol. This result could be justified by the WHO 2013 guidelines which recommend a first-line treatment with two NRTIs and one NNRTI before switching to a second-line treatment in case of virologic failure [25].

The median [IQR] duration of ART for the overall participants in this study was 7 [3–10] years, i.e., 5 [3–6] years in children and 7 [4–11] years in adolescents. According to the median duration of antiretroviral treatment, viral suppression was 58.21% (78/134) and 50.72% (70/138), respectively, in participants <7 years and ≥7 years on treatment. In fact, our results show that participants in this study are much less likely to achieve good virological success despite longevity on ART. Lack of adherence and of provision of adequate psychological support in children and adolescents has been reported to represent the main cause of loss to follow-up and virologic failure [26,27]. Non-adherence could be due to supply disruptions, more limited access to paediatric antiretroviral molecules or difficulties with therapeutic formulation [5,15]. Our results are very low compared with the study conducted in children by Teasdale et al., who found 84.0% viral suppression after two years of antiretroviral treatment, and the study conducted in adolescents by Fokam et al., who found 71% viral suppression after seven years of antiretroviral treatment [8,14].

The median (IQR) CD4 count of participants in this study was 576 (301–893) cells/µL, with 1113 (598–1505) cells/µL in children aged 1 to 5 years and 566 (295–855) cells/µL in children aged >5 years and adolescents. Immunological failure (defined as a CD4 count <500 cells/µL in children aged 1–5 years and <250 cells/µL in children aged 6–9 years and adolescents [28]) was 22.43% (61/272) overall, or 15.79% (3/19) versus 22.92% (58/253) in children aged 1–5 years versus children aged >5 years, respectively. Indeed, younger children have a better immune response on ART than older children [29]. This may be due to higher thymic production in younger children than in older children, or to greater damage to the CD4 T cell population in older children who have lived with untreated HIV for a prolonged period [18,30]. Our results are similar to the study by Teasdale et al. who found a median (IQR) of 565 (308–1138) cells/µL and the study by Fokam et al. who found a frequency of 21.38% in adolescents living with HIV with CD4 counts <500 cells/µL [8,14].

In this study, we did not find a statistically significant difference in gender disparities, although immunological failure was slightly lower in girls (21.43%, 27/126) than in boys (23.29%, 34/146). This may be explained by a better immune recovery observed in girls than in boys, which is in part due to greater compliance to antiretroviral treatment. This evidence is in accordance with our previous findings in a clinical setting in Cameroon wherein better immune recovery was found in female patients compared to males [8].

In this study, immunological failure in virally suppressed participants was very low (less than 10%) while it was extremely high (40%) in non-virally suppressed peers, underscoring the fact that restoration of immune function is better in the frame of virological control. In contrast, some in the paediatric studies have described immune recovery despite the absence of viral suppression, suggesting events of non-adherence or viral rebounds [31,32,33,34].

Thus, an in-depth assessment of children’s and adolescent’s adherence to treatment might help in strengthening therapeutic strategies in resource-limited settings. This calls for further studies to delineate optimal strategies to manage children/adolescents in advanced disease conditions. Additionally, further investigations are needed to identify children/adolescents with sustained viral control; the latter being eligible for antiretroviral treatment interruption toward a functional cure.

## 5. Conclusions

Conclusively, a low rate of viral suppression remains a major challenge in achieving the elimination of AIDS in paediatric populations within this sub-Saharan African setting. The consistency of poor virological response with poor immune status underscores delayed detection of ART failure, especially among those living in rural areas and those receiving NNRTI-based first-line ART. Henceforth, phasing out NNRTI-based regimens and close monitoring, especially among male children/adolescents in rural settings, would ensure optimal outcomes of paediatric ART in Cameroon and other resource-limited settings sharing similar features.

## Figures and Tables

**Table 1 tropicalmed-09-00048-t001:** Characteristics of the study population.

Variables	Total N = 272	Children (0–9 Years) N = 71	Adolescents (10–19 Years) N = 201
**Median age in years (interquartile range)**	13 (9–15.5)	7 (5–8)	14 (12–16)
**Gender**			
Female	126 (46.32)	34 (47.89)	92 (45.77)
Male	146 (53.68)	37 (52.11)	109 (54.23)
**Geographic location**			
Rural	42 (15.44)	7 (9.86)	35 (17.41)
Urban	230 (84.56)	64 (90.14)	166 (82.59)
**Health training**			
Private	18 (6.62)	7 (9.86)	11 (5.47)
Public	254 (93.38)	64 (90.14)	190 (94.53)
**Diagnostic Timing**			
Other	12 (4.41)	3 (4.23)	9 (4.48)
Consultation	93 (34.19)	17 (23.94)	76 (37.81)
PTME	167 (61.4)	51 (71.83)	116 (57.71)
**WHO clinical stages**			
I/II	248 (91.17)	63 (88.73)	185 (92.04)
III/IV	24 (8.83)	8 (11.27)	16 (7.96)
**Therapeutic line**			
1st line	213 (78.3)	55 (77.46)	158 (78.61)
2nd line	59 (21.69)	16 (22.54)	43 (21.39)

**Table 2 tropicalmed-09-00048-t002:** (**a**) Viral load distribution. (**b**) Determinants of Viral suppression.

**(a)**			
**Viral Load in Copies/mL**	**Children (%)**	**Adolescents (%)**	
<40	19 (26.76)	58 (28.86)	
40–999	22 (30.99)	49 (24.38)	
>999	30 (42.25)	94 (46.77)	
Total	71	201	
**(b)**			
	Viral suppression (<1000 copies/mL)		*p*-value
	Yes 148 (%)	No 124 (%)	
Age in years			0.60
1–9	41 (57.75)	30 (42.25)	
10–19	107 (53.23)	94 (46.77)	
Sex			0.33
Female	73 (57.94)	53 (42.06)	
Male	75 (51.37)	71 (48.63)	
Geographic location			0.04
Rural	17(40.48)	25 (59.52)	
Urban	131 (56.96)	99 (43.04)	
Therapeutic line			0.28
1st (NNRTI-based)	120 (56.34)	93 (43.66)	
2nd (PI/r-based)	28 (47.46)	31 (52.54)	
Duration of treatment (year)			0.26
<7	78 (58.21)	56 (41.79)	
≥7	70 (50.72)	68 (49.28)	

Legend. NNRTI: non-nucleoside reverse-transcriptase inhibitor; PI/r: ritonavir-boosted protease inhibitor.

**Table 3 tropicalmed-09-00048-t003:** (**a**) Distribution of CD4 counts in children aged 1 to 5 years. (**b**) Distribution of CD4 counts in children over 5 years of age. (**c**) Determinants of immunological failure.

(**a**)			
**CD4 Counts in Cells/µL**	**Number**	**Percentage (%)**	
<500	3	15.79	
500–999	6	31.58	
≥1000	10	52.63	
**Total**	19	100	
(**b**)			
CD4 counts in cells/µL	Number	Percentage (%)	
<250	58	22.92	
250–499	47	18.58	
≥500	148	58.50	
**Total**	253	100	
(**c**)			
	immunological failure		*p*-value
	Yes 61 (%)	No 211 (%)	
Age in years			0.66
1–5 *	3 (15.79)	16 (84.21)	
6–19 **	58 (22.92)	195 (77.08)	
Geographic location			0.10
Rural	14 (33.33)	28 (66.67)	
Urban	47 (20.43)	183 (79.57)	
Therapeutic line			0.01
1st (NNRTI-based)	55 (25.82)	158 (74.18)	
2nd (PI/r-based)	6 (10.17)	53 (89.83)	
Viral load in copies/mL			<0.0001
<1000	11 (7.43)	137 (92.57)	
≥1000	50 (40.32)	74 (59.68)	

Legend. *: immunological failure (CD4 < 500 cells/µL in the under-5 age group); **: immunological failure (CD4 < 250 cells/µL in the age group over 5 years). NNRTI: non-nucleoside reverse-transcriptase inhibitor; PI/r: ritonavir-boosted protease inhibitor.

## Data Availability

The datasets used and/or analysed during the current study are available from the corresponding author on reasonable request.

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
