# Peer review of "Determinants of Immunovirological Response among Children and Adolescents Living with HIV-1 in the Central Region of Cameroon"

_tropicalmed, 2024, doi:10.3390/tropicalmed9020048_

Round 1

Reviewer 1 Report

Comments and Suggestions for Authors

Line 56 – Please make clear that, while HIV viral load responds immediately to ART, dropping to undetectable typically in a few weeks, in the case of CD4 count the response is slower. If CD4 is low when starting ART, it may take months or more for it to climb to 500.

Line 98 – Please explain all the 3 UNAIDS goals, not just one of them.

Line 208 – Since you are describing median values, the numbers you write in brackets are probably the interquartile ranges (IQR). If so, please write it.

Tables 1 to 3 – You should also list the disease stage of the patients, for example, by referring to the WHO or CDC stages, or alternatively specifying how many had currently AIDS and how many already had AIDS in the past.

Discussion. You should say more about the symptoms and clinical history of the patients. For example, how many did have AIDS in the past. Also, what was the CD4 count when they started ART. If some patients started ART when CD4 was <200 (which is typical of patients with immunosuppression) they may have recovered, CD4 already raised, but it did not arrive yet to 500. Such cases should not be called immunologic failures, they display the normal CD4 behaviour. You should match these data with clinical history and also past CD4 to discern which patients have real immunologic failure.

Comments on the Quality of English Language

Line 5 – Some of the names are all in uppercase, others not, please be consistent.

Line 12 – “Yaoundé” has accents sometimes and sometimes not.

Line 16 – The affiliation number 6 from University of Rome is not listed in the author’s list. If someone from the University of Rome contributed to this manuscript this person should be listed as author.

Line 29 – Better “VS was achieved in 54.41%”.

Line 33 – “virally”.

Line 62 – You already defined HIV and AIDS, so why do you write it in extended form so many times in this paragraph?

Line 65 – “persons”.

Line 79 – The reports are called Demographic and Health Surveys, I checked in their website and it is still this name.

Line 89 – You already defined ART, so why do you write it in extended form so many times?

Line 96 and others – Again, you already defined HIV, AIDS, ART and UNAIDS, please use the acronyms.

Line 109 – “HIV early infant diagnosis” is not a correct expression. Do you mean early HIV diagnosis in infants?

Line 112 – You did not define “HIVDR” yet, please do so.

Line 122 – You can delete “which”.

Line 135 – You keep writing the extend forms and not the acronyms.

Line 186 – “Excel”.

Line 219 – Use the acronyms NNRTI and PI after you have defined them.

Line 270 – “Cameroon”.

Line 280 ­– Delete “in this study”, you already wrote “In our study”.

Line 283 – “World Health Organization”.

Line 312 – “higher”.

Line 317 – You can use the acronyms NRTI, NNRTI and PI throughout this paragraph (and in others).

Line 337 – “anti-retroviral”.

Line 377 – “in the pediatric”.

Line 380 – “among”.

Throughout the paper – better “virologic” than “virological” when referring to  virologic failure.

Author Response

Reviewer 1:

Comments and Suggestions for Authors

Comment 1: Line 56 – Please make clear that, while HIV viral load responds immediately to ART, dropping to undetectable typically in a few weeks, in the case of CD4 count the response is slower. If CD4 is low when starting ART, it may take months or more for it to climb to 500.

 Response 1: We thank the reviewer for this comment. We have added the clarification as suggested (see lines 61-64; page 2).

Comment 2: Line 98 – Please explain all the 3 UNAIDS goals, not just one of them.

 Response 2: We thank the reviewer for this comment. We have revised as recommended (see lines 105-112; page 3)

Comment 3: Line 208 – Since you are describing median values, the numbers you write in brackets are probably the interquartile ranges (IQR). If so, please write it.

 Response 3: We thank the reviewer for this comment. We have revised accordingly as recommended (see line 223; page 5)

Comment 4: Tables 1 to 3 – You should also list the disease stage of the patients, for example, by referring to the WHO or CDC stages, or alternatively specifying how many had currently AIDS and how many already had AIDS in the past.

Response 4: We thank the reviewer for this important comment. WHO clinical stages of children/adolescents enrolled have been added in table 1 (see lines 243-244, page 6). We did not collect data on their past clinical condition.

Comment 5: Discussion. You should say more about the symptoms and clinical history of the patients. For example, how many did have AIDS in the past. Also, what was the CD4 count when they started ART. If some patients started ART when CD4 was <200 (which is typical of patients with immunosuppression) they may have recovered, CD4 already raised, but it did not arrive yet to 500. Such cases should not be called immunologic failures, they display the normal CD4 behaviour. You should match these data with clinical

Response 5: We thank the reviewer for this comment. Unfortunately, patients’ files did not have all information required. We did not have CD4 counts at ART initiation, and reported only CD4 counts performed at enrolment in the study.

Comments on the Quality of English Language

Comment 6: Line 5 – Some of the names are all in uppercase, others not, please be consistent.

Response 6: We thank the reviewer for this comment. We have revised accordingly as recommended (see lines 5-10; page 1)

Comment 7: Line 12 – “Yaoundé” has accents sometimes and sometimes not.

Response7: We thank the reviewer for this comment. We have revised accordingly throughout the manuscript.

Comment 8: Line 16 – The affiliation number 6 from University of Rome is not listed in the author’s list. If someone from the University of Rome contributed to this manuscript this person should be listed as author.

Response8: We thank the reviewer for this comment. We have revised as recommended (see line 8; page 1)

Comment 9: Line 29 – Better “VS was achieved in 54.41%”.

Response 9: We thank the reviewer for this comment. We have revised as recommended (see line 31; page 1)

Comment 10: Line 33 – “virally”. 

Response10: We thank the reviewer for this comment. We have revised as recommended (see line 36; page 1)

Comment 11: Line 62 – You already defined HIV and AIDS, so why do you write it in extended form so many times in this paragraph?

Response11: We thank the reviewer for this comment. We have revised accordingly throughout the manuscript.

Comment 12: Line 65 – “persons”. 

Response 12: We thank the reviewer for this comment. We have revised as recommended (see line 72; pages 2)

Comment 13: Line 79 – The reports are called Demographic and Health Surveys, I checked in their website and it is still this name.

Response13: We thank the reviewer for this comment. We have revised as recommended (see lines 86–87; pages 2)

Comment 14: Line 89 – You already defined ART, so why do you write it in extended form so many times?

Response 14: We thank the reviewer for this comment. We have revised accordingly throughout the manuscript.

Comment 15: Line 96 and others – Again, you already defined HIV, AIDS, ART and UNAIDS, please use the acronyms.

Response 15: We thank the reviewer for this comment. We have revised accordingly throughout the manuscript.

Comment 16: Line 109 – “HIV early infant diagnosis” is not a correct expression. Do you mean early HIV diagnosis in infants?

 Response 16: We thank the reviewer for this comment. We have revised as recommended (see line 120; pages 3)

Comment 17: Line 112 – You did not define “HIVDR” yet, please do so.

Response 17: We thank the reviewer for this comment. We have revised as recommended (see line 125; page 3)

Comment 18: 122 – You can delete “which”.

 Response 18: We thank the reviewer for this comment. We have revised as recommended (see lines136; page 3)

Comment 19: Line 135 – You keep writing the extend forms and not the acronyms.

Response 19: We thank the reviewer for this comment. We have revised accordingly throughout the manuscript.

Comment 20: Line 186 – “Excel”.

  Response 20: We thank the reviewer for this comment. We have revised as recommended (see line 201; page 5)

Comment 21: Line 219 – Use the acronyms NNRTI and PI after you have defined them.

 Response 21:  We thank the reviewer for this comment. We have revised accordingly throughout the manuscript.

Comment 22: Line 270 – “Cameroon”.

  Response 22: We thank the reviewer for this comment. We have revised as recommended (see line 290; pages 9)

Comment 23: Line 280 ­– Delete “in this study”, you already wrote “In our study”.

Response 23: We thank the reviewer for this comment. We have revised as recommended (see line 301; page 9)

Comment24: Line 283 – “World Health Organization”.

Response24: We thank the reviewer for this comment. We have revised as recommended (see lines 304 and 306; page 9)

Comment25: Line 312 – “higher”.

Response25: We thank the reviewer for this comment. We have revised as recommended (see lines 345; page 10) 

Comment26: Line 317 – You can use the acronyms NRTI, NNRTI and PI throughout this paragraph (and in others).

Response26: We thank the reviewer for this comment. We have revised accordingly throughout the manuscript. 

Comment27: Line 337 – “anti-retroviral”.

Response27: We thank the reviewer for this comment. We have revised as recommended (see line 378; page 10) 

Comment28: Line 377 – “in the pediatric”.

 Response28: We thank the reviewer for this comment. We have revised as recommended (see line 414; page 11) 

Comment29: Line 380 – “among”.

Response29: We thank the reviewer for this comment. We have revised as recommended (see line 428; page 11) 

Comment30: Throughout the paper – better “virologic” than “virological” when referring to virologic failure.

Response30: We thank the reviewer for this comment. We have revised accordingly throughout the manuscript. 

Reviewer 2 Report

Comments and Suggestions for Authors

My only suggestion to the authors is to improve the discussion section.   I feel that there is a need to provide more specific information regarding two important issues:

a) More concise explanation of factors related to gender difference in virologic suppression and urban versus rural settings.    

b) Provide specific suggestions on improving virologic suppression using the same framework:  gender and setting (urban versus rural). 

Author Response

Reviewer 2:

Comments and Suggestions for Authors

My only suggestion to the authors is to improve the discussion section. I feel that there is a need to provide more specific information regarding two important issues:

Comment 1: More concise explanation of factors related to gender difference in virologic suppression and urban versus rural settings.   

Response 1: We thank the reviewer for this pertinent remark. We have revised the discussion section as recommended; providing more explanation as recommended (see lines 331-334 and 350-362; pages 9-10). 

Comment 2: Provide specific suggestions on improving virologic suppression using the same framework: gender and setting (urban versus rural). 

Response 2: We thank the reviewer for this comment. We have provided specific suggestions on improving virologic suppression as recommended (see lines 429-433; page 11).

Reviewer 3 Report

Comments and Suggestions for Authors

            The authors analyzed the determinants of immunovirological response to HAART in 272 children and adolescent living with HIV-1 in Cameroon. They found a higher prevalence of viral suppression failure in children from rural areas and a high relationship between failure of VS and immunological restauration as expected. Some concerns should be addressed before publication. Particularly, information about adherence should be provided for acceptance of this manuscript.

1.       The title could be changed to ¨among children and adolescents living with HIV-1¨.

2.       The Introduction is somehow long: the first paragraph description of the virus) could be reduced, and instead, include at the end of the Introduction the aim of this study.

3.       Page 3, Data collection: no information was requested on adherence to treatment? Adherence is crucial to prevent resistance and seems to have played a role, particularly in rural settings. This information is really missing in this study. The authors should make an effort to provide information on adherence, at least in a subgroup of the patients.

4.       Pages3-4, Quantification of plasma HIV-1 (please add 1, unless the kit also quantify HIV-2) viral load. There is no need to include all the details of the RT-PCR, just include according to manufacturer instructions.

5.       Page 4, CD4 lymphocyte count: same comment as 4.

6.       Was any information available on drug resistance in VS patients?

7.       Overall, the number of individuals tested is relatively small and this study should benefit of being accompanied on information of treatment adherence.

Author Response

Reviewer 3:

Comments and Suggestions for Authors

            The authors analyzed the determinants of immunovirological response to HAART in 272 children and adolescent living with HIV-1 in Cameroon. They found a higher prevalence of viral suppression failure in children from rural areas and a high relationship between failure of VS and immunological restauration as expected. Some concerns should be addressed before publication. Particularly, information about adherence should be provided for acceptance of this manuscript.

 Comment 1: The title could be changed to ¨among children and adolescents living with HIV-1.

Response 1: We thank the reviewer for this suggestion. We have revised the title just as suggested (see lines 2-3, page 1).

Comment 2: The Introduction is somehow long: the first paragraph description of the virus) could be reduced, and instead, include at the end of the Introduction the aim of this study.

Response 2: We thank the reviewer for this comment. We have revised the Introduction section as suggested and added the aim of the study at the end (see lines 42-110; pages 1-3).

       Comment 3: Page 3, Data collection: no information was requested on adherence to treatment? Adherence is crucial to prevent resistance and seems to have played a role, particularly in rural settings. This information is really missing in this study. The authors should make an effort to provide information on adherence, at least in a subgroup of the patients.

Response 3: We thank the reviewer for this relevant comment. In effect, adherence was routinely measured by self-reporting, and all participants reported being adherent, defined as haven received 95% or above of prescribed antiretrovirals. We have revised our result/discussion sections to include this crucial information (see lines 232-234 and 315-324; pages 5 and 9).

Comment 4: Pages3-4, Quantification of plasma HIV-1 (please add 1, unless the kit also quantify HIV-2) viral load. There is no need to include all the details of the RT-PCR, just include according to manufacturer instructions.

Response 4: We thank the reviewer for this comment. We have revised as recommended (see lines 153–156; pages 4). As for the procedure, the details were included for transparency purposes.

Comment 5: Page 4, CD4 lymphocyte count: same comment as 4.

Response 5: We thank the reviewer for this comment. All details were included for transparency purposes.

Comment 6: Was any information available on drug resistance in VS patients?

Response 6: We thank the reviewer for this comment. We acknowledge the fact that information on HIVDR will be of great relevance in monitoring treatment effectiveness. However, our dataset does not include information on drug resistance, as this was not a focus in our study objectives.

Comment 7: Overall, the number of individuals tested is relatively small and this study should benefit of being accompanied on information of treatment adherence.

Response7: We thank the reviewer for this comment. As aforementioned, adherence was routinely measured by self-reporting, and all participants reported being adherent. This was included in the discussion (see 315-324; page 9).

Round 2

Reviewer 3 Report

Comments and Suggestions for Authors

The authors addressed satisfactorely most of the reviewer´s comments.